# Comparing Barriers and Facilitators to Physical ActivityAmong Underrepresented Minorities: Preliminary Outcomes from a Mixed-Methods Study

**DOI:** 10.3390/ijerph22020234

**Published:** 2025-02-06

**Authors:** Rafael A. Alamilla, Navin Kaushal, Silvia M. Bigatti, NiCole R. Keith

**Affiliations:** 1School of Health and Human Sciences, Indiana University Indianapolis, Indianapolis, IN 46202, USA; nkaushal@iu.edu; 2Richard M. Fairbanks School of Public Health, Indiana University Indianapolis, Indianapolis, IN 46202, USA; sbigatti@iu.edu; 3School of Public Health—Bloomington, Indiana University Bloomington, Bloomington, IN 47405, USA; 4Center for Aging Research, Indiana University School of Medicine, Indianapolis, IN 46202, USA; 5Regenstrief Institute, Indianapolis, IN 46202, USA

**Keywords:** physical activity, barriers and facilitators, racial minorities, health disparities, mixed methods

## Abstract

Physical activity (PA)’s benefits are well established, yet many U.S. adults fail to meet PA guidelines. This is especially true for minorities facing social inequities. This study explored PA’s barriers and facilitators among urban Midwestern minorities using a mixed-methods approach framed on the socio-ecological model. A cross-sectional survey was conducted between January and June 2024 among community-dwelling minorities. Participants were grouped as completing low (LLPA) or high (HLPA) weekly leisure-time PA for comparison. Quantitative analysis included MANOVA, follow-up ANOVAs, and calculation of effect sizes. Qualitative data were assessed using inductive thematic analysis. Twenty-nine adults (44.83% Black, 41.37% Latino) participated in the study. The HLPA group (n = 18) reported higher leisure-time PA (*p* = 0.001, *d* = 2.21) and total PA (*p* = 0.02, *d* = 1.00) compared to the LLPA group (n = 11). LLPA participants faced more personal barriers to PA (*p* = 0.02, *d* = −0.92). Common barriers identified in the interviews included a lack of time and financial costs. Facilitators included social support and available PA facilities. Both groups achieved the USPA guidelines through different PA domains. Increasing social support and lowering PA-related costs could enhance participation. Addressing barriers and leveraging existing facilitators are crucial to increasing PA among minorities.

## 1. Introduction

### 1.1. Physical Activity and Implications for General Health

Physical activity (PA) is defined as any bodily movement that results in energy expenditure beyond a resting metabolic rate [1]. PA can be categorized into four distinct categories: occupational (i.e., working a physical labor job), transportation (e.g., bicycling to work), household (e.g., gardening), and leisure time (e.g., dog walking, exercising) [1,2]. Data on the health benefits of PA, specifically related to leisure-time PA, have been synthesized into what are now the United States PA (USPA) Guidelines for Americans [3] and the World Health Organization (WHO) Guidelines for PA and Sedentary Behavior [2]. Both guidelines recommend that adults aged ≥18 years should participate in at least 150 min/week of aerobic moderate-to-vigorous PA (MVPA) and engage in muscle-strengthening activities that target all major muscle groups at least 2×/week. These guidelines have been deemed attainable by most members of our society, including older adults, those with chronic health diseases, and individuals with physical or cognitive disabilities [2,3].

Data over the last two decades have suggested that meeting the USPA guidelines can promote lifelong health [4]. Moreover, regular PA is associated with many health benefits, including a reduced risk of developing chronic diseases and premature mortality, as well as improved mental health [2,5]. Evidence also suggests a dose-dependent relationship between PA and health, indicating that individuals who engage in higher amounts of PA, regardless of intensity, are more likely to have a lower risk of chronic disease [4] and improved overall health status [6].

Despite the well-known health benefits of PA, participation rates across the United States remain low [7,8]. Currently only 1 in 4 adults in the United States meets the PA guidelines for aerobic and muscle-strengthening activities [9]. These low PA participation rates are particularly concerning when considering the implications of physical inactivity for general health. Physical inactivity is considered to be the fourth-leading cause of all-cause mortality [10] and is a primary risk factor for chronic illnesses such as obesity, type 2 diabetes, cardiovascular disease, and cancer [10,11,12]. Considering the significant health implications of physical inactivity, efforts to promote PA are crucial to ensure that individuals and communities can sustain long-term health, increase their quality of life, and reduce the risk of illness and disability.

### 1.2. Health Disparities Faced by Underrepresented Racial Minorities

Data collected over recent decades suggest clear PA disparities between non-Hispanic Whites and underrepresented racial minorities (e.g., Black, Latino; hereafter “minorities”). Data have consistently shown that minorities are less likely to participate in regular PA than non-Hispanic White adults, regardless of the geographic region where they live [13,14,15]. To contextualize the PA disparities faced by minorities in the United States, it is essential to examine the social determinants of health (SDOH) that underpin these disparities. The SDOH refer to the conditions in an individual’s external environment that directly and indirectly affect their physical, mental, and emotional health [16]. These conditions have been shown to contribute to health disparities and include factors such as local/state economic conditions, education, health literacy, access to healthcare, and neighborhood environment [17].

Data suggest that minorities face considerable educational, economic, and social disparities that negatively affect their health outcomes [13,18]. More recent studies have shown that minorities are at a higher risk of contracting diseases such as COVID-19 [19] and have a higher incidence of preventable chronic illnesses [20,21,22]. Additional conditions that contribute to disparities include the built environment, which is defined as the human-made buildings and infrastructure that provide physical settings for individuals to live, work, learn, and engage in recreational activities [23]. Studies exploring the built environment have reported that minorities are more likely to live in communities with poorer walkability and limited access to resources [24,25,26], consequently resulting in worse health outcomes. The impacts of environmental determinants on PA extend beyond just the built environment, encompassing other environmental determinants such as air pollution and extreme weather [27]. Previous work has demonstrated an association between lower rates of PA participation among minorities and both poor air quality [6] and inclement weather [28,29]. Educational disparities are also faced by minorities, whereby these populations have been shown to have fewer opportunities to learn about healthy behaviors, such as PA [30,31]. Lastly, culture can have a significant influence on how individuals and communities engage in health behaviors, their environment, and larger systems that shape health [32,33]. In the case of PA participation among minorities, work has shown that cultural norms can influence their perceptions and participation [28,34,35]. Together, these data demonstrate that minorities face numerous disparities that make it difficult to adopt healthy behaviors. It is also evident that SDOH play direct and indirect roles in the adoption of PA among minorities [36]. These data emphasize the need for investigators to account for such factors when promoting PA in these populations, and to ensure that PA programs are constructed in a way that addresses the inequities faced by minorities.

### 1.3. Barriers to and Facilitators of Physical Activity Among Underrepresented Racial Minorities

Considering the disparities in regular PA participation faced by minority adults, it is evident that more work is required to facilitate opportunities and improve access to resources, which would support the adoption of PA behaviors for these populations. One way to start this process is to identify the barriers to and facilitators of PA experienced by minority groups. To date, considerable work has been conducted to describe PA’s barriers and facilitators in numerous populations with physiological conditions, such as individuals with cancer [37,38,39], diabetics [40,41], and adults with disabilities [42]. Extensive studies have also been conducted to describe PA’s barriers and facilitators using age as the primary descriptive variable, including children [43,44], young adults [45,46], and older adults [47,48,49]. However, race or ethnicity is seldom reported in large, longitudinal, population-based adult studies in this body of literature [50]. This lack of race or ethnicity reporting consequently requires more attention from investigators.

Recent research has begun to elucidate the barriers to and facilitators of physical activity among Latino populations in the United States. Larson and colleagues [34] reported that barriers to PA for Latinas were linked to cultural norms around caregiving and cultural standards for body shape. Additional work in this population has reported that Latinas receive little social support for PA, despite having large, close-knit social networks [28]. This is confounded by the presence of various environmental barriers, such as crime, extreme temperatures, traffic, lack of facilities to engage in PA, and fear of immigration enforcement [28,51,52]. Facilitators of PA in Latina populations have been shown to be similar to those among non–Latina Whites, with most research demonstrating that high self-efficacy and social support are motivating factors for engaging in PA [34,52]. Less work has been carried out in this area with Latino men [34], but existing work suggests that they share similar individual- and community-level barriers and facilitators when compared to Latinas [53].

The literature examining the PA’s barriers and facilitators among Black adults in the United States has suggested that this population shares many of the same barriers with Latino community members. In an analysis of four national datasets, investigators demonstrated that Black adults were less likely to participate in PA if they had less education and a lower household income when compared to non-Hispanic White adults [54]. Other studies with a primary focus on older Black women have demonstrated that psychological constructs such as a lack of self-efficacy, social support, and the ability to self-regulate PA behaviors were barriers to PA participation [29,55,56]. Work in the same population has also shown that environmental factors, such as neighborhood resource allocation and safety [56,57,58], can inhibit PA participation. The facilitators of PA documented in Black adults include factors such as perceived health benefits, social support from loved ones, and enjoyment derived from PA [29,57].

The literature on barriers to and facilitators of PA participation outlined here suggests that minority populations face barriers to PA that align with various SDOH, including the built environment, educational attainment, and economic stability. However, several gaps remain in the literature, necessitating the present investigation. To date, data have primarily focused on minority groups situated in the Southwestern and Eastern portions of the United States, with minimal exploration of minorities in the Midwestern region of the United States. This region has numerous cultural, economic, and environmental differences from the aforementioned regions that make it distinct, which could alter the ways in which previously reported barriers and facilitators influence PA participation. The literature on PA’s barriers and facilitators has also underreported how these factors differ between minority adults completing low and high amounts of PA. Exploring these differences may provide investigators with nuanced information that can better inform how individuals within the same group perceive and experience PA. Lastly, studies have yet to fully elucidate how the long-term impacts of the COVID-19 pandemic have affected PA’s barriers and facilitators in minority populations. COVID-19 intensified existing disparities faced by minority populations [59], including access to and opportunities to engage in PA [60]. Given that numerous economic, social, and environmental factors stemming from the pandemic persist today, this study is crucial for elucidating how these external elements may be associated with current levels of participation in physical activity. Taken together, the present study aims to provide novel findings that will provide important information for individuals aiming to promote PA in this region of the United States.

### 1.4. Using the Socio-Ecological Model to Identify Barriers to and Facilitators of Physical Activity

Theoretical frameworks provide investigators with structured and reproducible methodological approaches for answering research questions [61]. The use of theoretical frameworks can facilitate stronger PA programs in community settings by introducing consistent, structured, and holistic approaches to implementing such programs. Framing PA programs using theoretical models can help investigators and community stakeholders (i.e., health officials, legislatures, etc.) to interpret program outcomes, better inform best practices for additional health programs, and provide stronger evidence for health-related policies. The present project explores PA’s barriers and facilitators among minorities using a socio-ecological model (SEM) [62]. The SEM suggests that health outcomes result from a complex interplay among individual, interpersonal, community, and societal domains that comprise an individual’s life. These domains encompass a wide range of factors, such as individual-level biology, interpersonal relationships, community resource availability, the built environment, and state/federal policies. It is important to note that this model suggests that factors in one domain can impact those in another domain, suggesting that it is necessary to account for all domains simultaneously to promote healthy behaviors.

The individual domain of the SEM encompasses factors related to an individual’s personal wellbeing, such as their psychological and physical states. Studies in the health behavior literature have demonstrated that various psychological constructs can play pivotal roles in PA participation. At the conscious level, work has shown that having an exercise identity (i.e., participating in PA is a key aspect of someone’s self-concept) can lead individuals to be more physically active [63]. Habit formation allows PA to be initiated with less conscious effort and reduces the mental burden necessary to continue PA in the long term [64,65]. Regarding the role of physical health, previous studies have suggested that individuals who perceive themselves as having physical limitations or a low level of physical fitness are less likely to engage in PA [48,66]. Together, these intrapersonal factors have been shown to play key roles in the adoption and maintenance of PA.

The interpersonal domain of the SEM can encompass the various social relationships an individual has and whether they feel socially supported to participate in a behavior. Social support refers to the perceived availability of social resources and support from an individual’s social relationships, comprising five distinct forms [67]: emotional (i.e., receiving encouragement), companionship (i.e., having a sense of belonging, having an activity partner), instrumental (i.e., having the resources needed to carry out a behavior), informational (i.e., having easily accessible information on how to perform a behavior), and validation (i.e., seeking others for social comparison, normative behavior). Social support has been shown to be a key factor that can help individuals remain physically active [68,69]. Conversely, lack of social support has been documented as a barrier to participation in regular PA [34]. These results suggest that social support plays a pivotal role in regular PA engagement.

The community domain of the SEM includes the social and physical settings where individuals live, learn, and work. The built environment is of particular interest to investigators and community stakeholders given its demonstrated direct impact on health outcomes [6]. Previous work has demonstrated that built environments possessing PA-supportive features (e.g., gyms and green spaces) can facilitate more PA [70,71] than communities that lack these attributes. Other neighborhood and built environment factors, such as crime rate, neighborhood walkability, and the prevalence of inclement weather [24,25,26,72], have also been shown to impact PA participation. Within the context of organizational environments (i.e., workplaces, college campuses, etc.), these entities can support PA by facilitating spaces that encourage PA. This may include providing easily accessible fitness facilities, subsidizing health insurance premiums, providing regular wellness workshops or onsite personal trainers, and other organization sponsored initiatives [73]. Prior work suggests that PA-supportive organizational policies can promote PA [74,75] and, in turn, provide an additional avenue to support positive health outcomes. Together, these data suggest a complex relationship between environmental factors and PA participation, underscoring the need to explore these factors when considering potential barriers to and facilitators of PA among minorities.

The societal domain of the SEM comprises state/federal policies, social norms, and cultural norms that broadly influence individuals and communities. Public policies that distribute funds and resources to communities can have direct downstream impacts on PA participation [18,31]. Policies that shape the built environment, such as Complete Streets [76] and neighborhood zoning laws [77], can also play pivotal roles in individual- and community-level PA trends by instituting pedestrian-friendly features as a community default. To further support this point, recent work suggests that community residents are more likely to have good health if their neighborhoods have implemented pedestrian-friendly policies [78]. Cultural and social norms have also been shown to influence PA participation, particularly among minority populations. Studies conducted in Black and Latino communities have suggested that cultural and social norms surrounding caregiving and prioritization of the family unit over individual pursuits [34,35] are barriers to PA. Cumulatively, this domain of the SEM emphasizes the need for multifaceted strategies to promote PA, which take into account both top–down policy measures and underlying cultural norms present in diverse communities.

### 1.5. Study Purpose

The primary purpose of this study was to explore and contextualize the barriers to and facilitators of PA faced by urban, Midwestern, minority adults. The specific aims of this investigation were (1) to qualitatively and quantitatively describe the existing barriers to and facilitators of PA participation among minority community members, and (2) to explore differences in PA’s barriers and facilitators between minorities completing high and low amounts of leisure-time PA. It was hypothesized that (1) minority adults would report barriers and facilitators that would span across all four domains of the SEM, and that (2) minorities completing low amounts of leisure-time PA would report more barriers to and fewer facilitators of PA.

## 2. Materials and Methods

### 2.1. Study Design

This project utilized a cross-sectional, concurrent mixed-methods [79,80] study design that aimed to contextualize PA’s barriers and facilitators among minority adults using the SEM. This manuscript follows the APA JARS Mixed-Methods Article Reporting Standards [81] guidelines.

### 2.2. Participant Recruitment and Eligibility Criteria

Community-dwelling minority adults were recruited from an urban Midwestern county between January and June 2024, using convenience and snowball sampling. Existing community partners within the target county aided in the dissemination of study recruitment materials via word of mouth, the promotion of recruitment flyers, and email correspondence with community members. The study staff also promoted recruitment materials via social media platforms, posts in public libraries, and tabling at local health-based events.

The inclusion criteria for the present study included (1) self-identification as a racial minority, (2) being a resident living within the specified geographic area, and (3) being ≥18 years old. The exclusion criteria included (1) not having access to a phone or computer, (2) being unable to read and speak English or Spanish fluently, and (3) being unable to provide written informed consent. The participants were screened by phone or video calls and provided written informed consent before enrolling in the study. The individuals did not have an existing relationship with the primary investigator at the time of enrollment, and they were made aware of the research team’s reasons for and interest in the research topic. The participants were compensated with a USD 25.00 gift card for completing the study.

### 2.3. Study Survey

The participants completed a 30–45 min electronic survey that aimed to explore potential barriers to and facilitators of PA across the various domains of the SEM. Upon obtaining informed consent, the participants were sent a link that directed them to the survey and instructed them to complete the survey in one sitting. The survey was completed on a personal computer or mobile device. A paper copy of the survey was made available upon request. All surveys were delivered by the principal investigator, who was trained in conducting mixed-methods research.

The study survey comprised various sets of previously validated questionnaires that aimed to assess current weekly PA levels, as well as factors such as PA habit formation, exercise self-identity, perceptions of social support, and influences of the physical environment. The survey was linguistically and culturally translated into Spanish for application to Spanish-speaking demographics of the surveyed area by a fluent Spanish-speaking member of the research team. A different fluent Spanish-speaking member of the research team then back-translated the translated survey, ensuring that all translated items maintained semantic, idiomatic, and grammatical equivalence compared to the original English versions. Finally, the survey was refined for comprehension and understanding through pilot testing with Spanish-speaking community members.

#### 2.3.1. International Physical Activity Questionnaire

The International Physical Activity Questionnaire-Long Form (IPAQ-LF) provides a set of well-developed survey questions that provide self-reported weekly estimates of PA [82]. The long-form questionnaire assesses weekly PA across the four established domains (occupational, transportational, household, and leisure-time PA,) with acceptable concurrent and construct validity [82] as well as good test–retest reliability [83]. The participants were asked to report the weekly frequency (i.e., days per week and minutes per day) of light, moderate, and vigorous intensity activities performed across each domain, which were then analyzed to determine weekly minutes.

#### 2.3.2. Barrier Analysis Survey

The Barrier Analysis Survey is a qualitative rapid assessment tool created to assess barriers to and facilitators of a target health behavior [84,85]. The survey includes a combination of standardized open- and closed-ended questions framed on the health belief model and the theory of reasoned action, as well as questions that assess participants’ perceptions related to access to resources, policy, and culture [84,85]. The set of standardized questions [85] was adapted to explore the determinants of behavior associated with participating in PA. All questions were presented as part of the survey and placed towards the beginning to account for participant response fatigue [86,87].

#### 2.3.3. Physical Activity Barrier Questionnaire

The Physical Activity Barrier Questionnaire (PABQ) is a 24-item questionnaire that assesses potential barriers to PA in adults aged >18 years. The questionnaire follows the SEM, wherein questions are categorized into three domains: personal, social environment, and physical environment [88]. All items are scored on a Likert scale ranging from 1 (“strongly disagree”) to 5 (“strongly agree”). Previous reporting has demonstrated this survey to have good internal consistency (α = 0.86) as well as strong face and content validity [88].

#### 2.3.4. Exercise Identity Scale

The Exercise Identity Scale (EIS) is a 9-item scale that assesses the salience of individuals identifying with exercise as an integral part of their self-concept [89]. The questions are delivered on a 5-point Likert scale and anchored by a stem that contextualizes each item within their personal exercise experience (e.g., “The following questions concern your personal beliefs about exercise. Please indicate the degree to which you agree or disagree with each statement when thinking about your exercise participation.”). Previous work has shown the EIS to have good internal reliability (α = 0.82–0.95) and criterion validity [90].

#### 2.3.5. Physical Activity and Social Support Scale

The Physical Activity and Social Support Scale (PASSS) is a 20-item scale that assesses the perceived presence of social support related to PA among adults [67]. The PASSS has been shown to measure the five forms of social support (i.e., companionship, emotional, informational, instrumental, and validation) with good internal reliability (α = 0.89) and acceptable discriminant and convergent validity. The survey items are prefaced with a statement regarding the different ways in which an individual can receive support before prompting respondents to indicate how well a statement relates to the PA they engage in on a 7-point Likert scale, with 1 = never true and 7 = always true [67].

#### 2.3.6. Self-Report Behavioral Automaticity Index

The Self-Report Behavioral Automaticity Index (SRBAI) is a validated 4-item instrument that has been shown to be a valid and internally reliable (α = 0.86) survey capable of characterizing the automaticity of a habit in adults [64]. The SRBAI’s items were slightly modified to assess the preparatory phase of habit [91] by adjusting the question stem (i.e., “When I prepare to exercise”), which was then followed by the four items on the scale. The items were presented on a 5-point Likert scale, with 1 = strongly disagree and 5 = strongly agree [64,91].

### 2.4. Participant Demographics

The participants were asked to respond to questions regarding their race/ethnicity, height/weight, income, education, the community they lived in, and how long they had been residents of their community.

### 2.5. Determination of Study Groups

Participants were labelled into one of two groups for analysis based on self-reported PA from the IPAQ-LF: Low Leisure Physical Activity (LLPA) or High Leisure Physical Activity (HLPA). Individuals in the LLPA group were those who reported completing <150 min/week of leisure-time PA. Individuals in the HLPA group were those who reported completing >150 min/week of leisure-time PA. Leisure-time PA was calculated by summing the reported weekly minutes of leisure-time walking and exercise derived from the IPAQ-LF. The emphasis on leisure-time PA as a differentiator between groups is consistent with PA messaging focused on promoting leisure and exercise PA prominently within both the United States and WHO PA guidelines [2,3].

### 2.6. Data Analysis

#### 2.6.1. Sample Size

An *a priori* power analysis using G-Power [92] was performed to conduct a cross-sectional analysis with two groups and ten response variables. Power analysis revealed that a sample of 76 participants was needed to detect significant differences (*p* = 0.05) at a medium effect size (F^2^ = 0.25), with a power (1 − β error probability) set at 0.80 for all primary quantitative outcome measures [93]. Cohen’s *d* was used to identify the effect size differences between groups. The estimation of effect size differences provides an acceptable way to quantify the magnitude of the difference between group means. Effect size estimates were interpreted using guidance provided by Ferguson [94], which suggests that an effect size *d* of 0.41 or greater is the recommended minimum effect size necessary to achieve a practically significant effect.

#### 2.6.2. Quantitative Data Analysis

SPSS (version 29.0) was used to conduct all analyses, including the computation of descriptive and inferential statistics. Survey composite scores were calculated by first converting Likert scale items to numeric values and subsequently summing the item scores to obtain the subdomain and total survey scores. Prior to the hypothesis tests, Little’s Missing Completely at Random (MCAR) test [95] was performed to identify whether data were missing at random. If the criteria of MCAR and other data patterns were met, multiple imputations were used to estimate the missing data [96,97]. The primary hypotheses were tested by conducting a MANOVA to assess the multivariate effects of the independent variables (weekly leisure-time PA min/week) on the dependent variables (survey composite and sub-composite scores). Significant tests at the multivariate level were further explored using one-way ANOVA. Weekly amounts of PA by domain were tested by conducting a series of ANOVAs. Differences in demographic data were assessed by independent-samples *t*-tests. A *p*-value of *p* < 0.05 was considered statistically significant. Cohen’s *d* was computed for between-group differences among study groups for all primary outcomes [93,94].

#### 2.6.3. Qualitative Data Analysis

Participants’ responses to the standardized set of open-ended questions from the Barrier Analysis Survey were transferred to an Excel (version 2412, Redmond, Washington DC, USA) sheet verbatim and verified by the primary investigator. Themes and sub-themes were derived from the data using inductive thematic analysis [98], a flexible method that seeks to understand participants’ experiences, thoughts, and behaviors through the active construction of themes derived from a dataset. This analysis approach comprises a six-step process that involves (1) familiarization with the data, which was achieved by reading and re-reading participants’ responses; (2) establishing coding rules and generating initial codes; (3) searching for initial themes; (4) reviewing and refining the initial themes; (5) defining and naming themes and sub-themes; and (6) producing a final report [98].

Multiple strategies were undertaken to ensure the credibility, dependability, confirmability, and transferability of these data. Two members of the study team analyzed the transcripts separately and then came together to discuss their interpretations of the data. The inter-rater reliability agreement threshold was set to ≥70%. A consensus among the research team personnel was reached for all initial codes before the subsequent coding and generation of themes commenced. During all discussions, a “critical friends” approach was undertaken [99]. In this approach, critical dialogue between members of the research team was undertaken, whereby members gave voice to their interpretations of the data in relation to other study staff, who listened and offered critical feedback. The aim of this approach was to encourage reflexivity and provide a theoretical sounding board to encourage the exploration of multiple alternative explanations and interpretations of the data. This approach has been shown to be acceptable for demonstrating rigor in analyzing qualitative data [99,100]. The participants did not provide feedback on the qualitative findings at this stage of the study. Data were reported in accordance with the Consolidated Criteria for Reporting Qualitative Research (COREQ) guidelines [101].

## 3. Results

After screening to ensure that the individuals met the study’s inclusion criteria, 29 participants completed the survey (LLPA, n = 11; HLPA, n = 18). The majority of the participants were female (86.21%) and educated (65.52% held a bachelor’s degree or more); 44.83% identified themselves as Black, and 41.37% identified themselves as Latino. Independent-samples t-tests did not reveal significant differences in demographic variables (*p* > 0.05). Participants in the HLPA group engaged in more leisure-time PA (LLPA = 44.55 ± 48.40 min/week, HLPA = 448.89 ± 227.43 min/week; F_(1, 27)_ = 33.39, *p* = 0.001, *d* = 2.21) at the time of enrollment. Practical effect size differences were observed between groups for body mass (*d* = −0.53) and body mass index (BMI, *d* = −0.42). The remaining demographic data can be found in Table 1. A full description of the participant flow through screening and informed consent is shown in Figure 1.

### 3.1. Missing Data Analysis

Missing data analysis revealed that 0.088% of the data were incomplete. Little’s MCAR test found the data to be missing completely at random (χ^2^ = 0.00, DF = 154, *p* = 1.00). Since the proportion of missing data observed was below 5.00%, missing data were ignored during analysis, and multiple imputations were not conducted [96].

### 3.2. Self-Reported Physical Activity

Full descriptions of the PA domain-specific means, standard deviations, percentage differences, and between-group differences can be found in Table 2. Participants in the HLPA group reported engaging in significantly more total PA (F_(1, 27)_ = 6.80, *p* = 0.02), total walking PA (F_(1, 27)_ = 6.91, *p* = 0.01), leisure-time walking PA (F_(1, 27)_ = 22.08, *p* < 0.001), and exercise MVPA (F_(1, 27)_ = 12.34, *p* = 0.002) than those in the LLPA group. While not significant, practical effect size differences were observed, favoring the HLPA group for occupational walking PA (*d* = 0.54), total occupational PA (*d* = 0.53), and total MVPA (*d* = 0.67).

### 3.3. Quantitative Survey Results

The MANOVA testing survey composite scores were found to be statistically non-significant between groups (F_(1, 27)_ = 1.18, *p* = 0.37; Wilk’s λ = 0.61) but yielded a practical effect size difference (partial η^2^ = 0.40) at the multivariate level [94]. Follow-up univariate tests revealed that the LLPA group faced significantly more personal barriers to PA than the HLPA group (F_(1, 27)_ = 5.72, *p* = 0.02, *d* = −0.92). The total composite scores for the PABQ were significantly different between groups (LLPA = 60.73 ± 19.90, HLPA = 47.44 ± 12.84; F_(1, 27)_ = 4.81, *p* = 0.04, *d* = −0.84). The total composite scores for the PASSS were not statistically significant, but they did achieve a practical effect size difference (LLPA = 59.00 ± 13.21, HLPA = 67.00 ± 20.31; F_(1, 27)_ = 1.35, *p* = 0.26, *d* = 0.44) [93,94]. Full descriptions of the survey composite score means, standard deviations, and between-group differences can be found in Table 3.

### 3.4. Qualitative Survey Results

Inductive thematic analysis revealed that the HLPA group had 11 distinct barriers to PA and 16 distinct facilitators of PA. The LLPA group had 11 distinct barriers to PA and 12 distinct PA facilitators. Analysis revealed that both groups shared six PA barriers and eight PA facilitators. Descriptions of prominent barriers and facilitators across each of the SEM domains are provided below. A full description of the identified barriers to PA for both groups, with associated quotes, can be found in Table 4. A full description of the identified facilitators of PA for both groups, with associated quotes, can be found in Table 5.

#### 3.4.1. Personal-Level Barriers

**Lack of Time**: Inductive thematic analysis revealed that both groups repeatedly mentioned that their work schedules made it difficult to find time to be active. One participant from the LLPA group mentioned that *“Balancing full time work and part time school [makes it hard for me to be active]”*, while another LLPA group participant mentioned that *“[Physical activity] takes time away from work, causing me to give up sleep, take an unpaid lunch break, and/or work longer hours”.* These quotes suggest that scheduling PA into daily and weekly schedules was an issue for these participants. While individuals in the HLPA group also reported work schedules as a factor that limited their time to be active, HLPA group members mentioned that school, family obligations, and other responsibilities prevented them from finding time to be active. One participant mentioned that *“School… discourages me from being active as [school] takes up too much time in my schedule”.* Another participant stated that *“Family obligations, work obligations, and lack of time [makes it hard to be active]”*. A third participant highlighted that having two jobs made it exceedingly difficult to be active, stating that *“Finding time [makes it hard to be active]. I have two jobs that take up every single day all day”.* Together, these quotes suggest that participants have various competing life demands that make it complicated to be physically active.

**Financial Costs Associated with PA:** Cost was a prevalent personal barrier for both study groups, as well as being an identified sub-theme within other primary themes within the community and societal domains. Several participants mentioned that engaging in PA was expensive and prohibited them from being active. One participant mentioned that *“expensive gym memberships [discourage me from being active]”*, while another participant mentioned *“programs that are expensive and not local [discourage me from being active]”.* When asked about what would help facilitate PA, participants from both groups mentioned that free or reduced-cost PA resources would help them be more active. Proposed free or reduced-cost resources included community PA classes (*“Free community offered classes, and physical events (yoga, dance, walking/running,) [would help me be more active]. A community gym offered at a free and or very reduced rate.”*), community walking programs (*“I’m already interested in anything free, but especially if I have an interest in an activity like yoga. A community walking program would be very helpful and fun to get into.”*), and free access to local gyms (*“Free gym memberships with a trainer [would help me be more active]”*).

#### 3.4.2. Personal-Level Facilitators

**Knowledge of PA Benefits**: A majority of participants across both groups reported knowing about the positive health benefits that can be achieved with regular PA. Two participants from the HLPA group reported that they knew about the linear relationship between the amount of PA one engages in and the amount of health benefits one can achieve—one stating that they were *“fairly knowledgeable [about the benefits of physical activity]. The more active you are, the better health you’ll have”.* Participants from both groups commented that they knew that PA had many benefits for their physical health, such as keeping their heart and joints healthy (*“Physical activity leads to good cardiac and joint mobility.”*). Participants also mentioned knowing about the link between PA and mental and emotional health. One participant stated the following regarding this link:


*“If you don’t use it, you will lose it. Staying active activates the happy hormone and relieves stress; it produces clarity of the mind… walking in nature produces some peace and grounding… Physical activity is providing a holistic measure for producing good health, mind, body, and spirit; good health is wealth”.*


**Health Benefits Accrued from PA:** Participants across both groups reported that PA has helped them gain health benefits in the past, and that these past experiences motivate them to try to continue being active. Inductive thematic analysis revealed that the mental health benefits accrued from PA were a prevalent sub-theme, with several participants stating that PA helped them improve their mood, enhance their confidence, and decrease depressive thoughts. One participant from the LLPA group stated that *“[physical activity] helps with sleep, mood, concentration, and balancing mental health struggles”.* A different participant from the HLPA group stated that PA helped give them a *“clear mind, good attitude, and decrease in depression”,* while another participant from the same group stated that PA provided them with *“better mental health, better mood, more energy and increase[d] productivity”.*

#### 3.4.3. Interpersonal-Level Facilitators

**Social Support:** The majority of participants from both study groups reported that they had individuals in their lives who directly or indirectly supported them in being physically active. When asked about who in their life supports them in being active, the participants mentioned that immediate family (i.e., parents, siblings, spouses, etc.) would encourage them. One participant mentioned that *“my husband, my mother [support me]”* to be active, while a different participant said, *“My fiancé, kids, and best friend [support me]”.* In addition to immediate family support, others mentioned that they had social support from extended social circles, including friends (*“Having a gym buddy [helps me stay active].”*), work colleagues (*“I do ‘Walking Wednesdays’ with colleagues.”*), and members of their community (*“My community consists of active older adults who are generally very physically active*). The general consensus among the participants was that having others discourage them from being active was uncommon. As one participant put it, *“I don’t think I have ever run into someone that has discouraged me from being active”.*

#### 3.4.4. Community- and Societal-Level Barriers

**Lack of PA Facilities or Transit Options:** One participant mentioned that *“[Not having] a convenient gym [makes it hard to be active]”*, while another participant stated that *“[Not having] easy access to public transport [makes it harder to be active]”.* Two individuals in the LLPA group stated that inclement weather was a barrier to being active, one of whom stated that *“Some days, the weather doesn’t allow for me to get outside and enjoy walking or driving to the local YMCA for group exercise”.*

#### 3.4.5. Community- and Societal-Level Facilitators

**Community PA Facility Availability:** Both groups were clear that easier access to PA facilities in their community would facilitate more PA. When asked what community resources would make it easier for them to be active, several participants stated that having access to a free gym (*“A good gym that was also free would help me be more active.”*) and having access to sports-specific facilities (*“More places to play tennis or pickleball [would help me be more active]”*) would facilitate PA for them. As mentioned, cost was a prominent sub-theme in many of the responses. When asked about what community resources would help them be more active, one participant responded that *“Free access to gym/ recreational activities and resources for beginners [would help me be more active]”.* A separate participant shared this sentiment, stating that *“Free or super reduced [gym] memberships [would help me be more active]”.*

**Community PA Events and Programs:** Inductive thematic analysis revealed that having more PA-related events and programs in their community would help the participants to be more active. Specifically, participants commented that regular pick-up sports games (*“regular cricket games in [the] softball field. Having some table tennis tournaments”*) and outdoor fitness programs (*“Free community offered classes, and physical events (yoga, dance, walking/running,)…”*) would facilitate PA for them. Having employers support PA through sponsored programs was also reported by participants as a facilitator to PA. As one participant commented when asked about what policies would support them in being active, *“Something with employment, money back for being active [would encourage me to be active]”*.

## 4. Discussion

The present study aimed to explore the existing barriers to and facilitators of PA among urban Midwestern minorities using a concurrent mixed-methods approach framed by the SEM. This study is the first to the authors’ knowledge that has intentionally compared PA’s barriers and facilitators among minorities completing low and high amounts of leisure-time PA. This study is also novel in that this investigation assessed PA’s barriers and facilitators using a theory-driven, mixed-methods approach among urban Midwestern minorities, a minority population that is underexplored in the current body of literature. Our results present novel findings that can inform future PA interventional programs.

The present study revealed that the HLPA group engaged in higher weekly leisure-time PA than the LLPA group and exceeded the USPA Guidelines [3], and we used the IPAQ-LF to document weekly PA performed by the study participants across all PA domains [1]. The IPAQ-LF is an easily administered, globally validated survey [83] that can estimate PA with moderate validity compared with accelerometry-derived PA estimates [102]. Documenting PA across all domains is important to help investigators to have a full understanding of someone’s weekly activity trends. Historically, estimates of PA have focused on activities performed almost exclusively during leisure time [103]. This tendency to measure only leisure-time PA may consequently result in the underestimation of an individual’s weekly PA levels [104].

The LLPA group achieved the USPA guidelines primarily through occupational and household MVPA. These outcomes are consistent with those of previous studies [104,105,106], demonstrating that minorities are more likely to be categorized as meeting the USPA guidelines when all domains are considered in the estimation of weekly PA. While these data are initially encouraging, recent studies suggest that not all PA domains have the same impact on health [107,108]. Work focusing on health outcomes stratified by the amount of occupational PA suggests the presence of a “PA health paradox” [107], whereby occupational PA has been shown to be associated with adverse impacts on cardiovascular health and all-cause mortality [109,110,111,112]. Given that minorities are more likely to work in more physically demanding occupations [113] and are less likely to engage in leisure-time PA [114], future work should aim to further document how PA across various domains impacts health outcomes in minorities, and to investigate how these trends influence individuals’ willingness to participate in research and community PA programs.

The quantitative outcomes from this study revealed that participants completing low amounts of leisure PA reported more personal barriers to PA. These findings were further supported by the qualitative outcomes, which reported more personal barriers in the LLPA group. Regarding psychological barriers, the LLPA group reported having less motivation and discipline to be active, as well as expressing a dislike for the physical sensations (i.e., soreness, sweating, etc.) associated with being active. These findings are consistent with previous studies [29,52] and lend further support to the presence of persistent personal barriers to PA in this population. The psychological barriers reported by the participants align with the affective determinants of PA [115]. These determinants, such as how someone feels when performing an activity, or having the motivation to engage in activity, have been shown to impact future PA behavior. Specifically, individuals who have positive associations between affective responses and PA are more likely to continue their participation [116,117].

The quantitative results from the current study also suggest that the HLPA group was more likely to report a stronger exercise identity and exercise behavior that was more habitual. While not explicitly stated in the participants’ quotes, several participants in the HLPA group alluded to the idea that PA was an important part of their lives, lending further support to the quantitative findings. To the best of the authors’ knowledge, this is the first investigation in the existing literature on this topic to consider these psychological constructs when exploring potential barriers to and facilitators of PA. Previous work has demonstrated a positive correlation between habit and PA participation [118], suggesting that having an existing habit can be predictive of future PA [65]. Numerous studies completed with new exercisers have shown that successfully building a PA habit can facilitate higher weekly PA [119,120]. In addition, having an exercise identity has been shown to help strengthen PA-related behaviors by positively impacting an individual’s self-concept [63]. Taken together, future PA programs should consider embedding strategies that positively reinforce these constructs to help minorities adopt regular PA.

The qualitative results from this study suggested that the LLPA group had lower perceived physical capacity and was more conscious about the injury risks associated with PA. These voiced perceptions were further supported by the individual PABQ personal domain item response rates, which suggested that the LLPA group had physical limitations that prevented PA participation. Low physical capacity and the risk of injury are both prevalent barriers to PA among minorities [29,34,121]. These perceptions are consistent with previous work, suggesting that individuals are less likely to engage in PA if they have an injury or physical limitation [48,66]. The USPA guidelines suggest that, assuming no contraindications are present, individuals of any ability or physical state can participate in PA. One approach to helping individuals build confidence to be active is to build and reinforce physical literacy, which refers to the competence, confidence, and knowledge required to be physically active in a variety of environments. Physical literacy has been established as a key determinant of health in adults, with preliminary evidence suggesting a positive relationship between physical literacy and health outcomes [122]. In turn, future work should aim to develop feasible approaches to enhance physical literacy in adults, so as to help individuals overcome their fear of injury and doubts of their physical capacity.

Responses to the qualitative survey questions revealed that the lack of time to be active was a prominent barrier for both of the study groups. This outcome is further supported by the individual PABQ social environment domain item response rates, which suggest that both groups had life demands that limited their time to be physically active. These data further support the previous literature reporting time to be a significant barrier to PA [28,29,34,49,121,123]. These findings are unsurprising considering that minorities are more likely to work long, irregular hours [124] that prevent the establishment of a PA habit, and to have different cultural norms around the family that may limit time for PA [28]. Finding strategies to help individuals accommodate these factors may, in turn, require a combination of different programmatic approaches, such as the implementation of action planning and adjusting how programs prescribe PA to participants. Action planning is a self-regulatory strategy that has been shown to bridge the intention–behavior gap by helping individuals to establish goals, plan for potential barriers, and improve their time management [125]. Action planning has been successfully implemented in new exercisers [119,126], and it may be a viable approach to help individuals find time to be active. In addition, PA program facilitators may consider adjusting program-facing activity recommendations to encourage shorter bouts of activity spread across the day (i.e., ~10 min bouts several times per day) over one longer bout per day (i.e., 3×/week for 45 min), in order to account for existing time constraints. This approach to PA engagement has been shown to lead to positive health benefits [127] and remains in line with the existing PA guidelines [3].

Financial cost appeared as both a primary theme and a sub-theme among the participants’ responses, suggesting that the participants had cost in mind when making decisions about being active. These qualitative data are further supported by the outcomes of the PABQ response rates to items related to cost, which suggest that more than half of the LLPA group participants and one-third of the HLPA group participants agreed that cost was a barrier to PA. Cost has repeatedly been reported as a barrier to PA participation [42,48,128,129]. This is largely unsurprising given the combination of existing disparities in wage earnings between minorities and non-Hispanic White adults [113], as well as existing sentiments around inflation and the cost of living in the United States [130]. Previous work carried out on minorities has attempted to address financial barriers by providing free or reduced-cost PA programs [131,132]; however, it is unclear to what extent these economic benefits continue for participants beyond the end of these interventions. Considering that economic stability is a key SDOH, future PA programs should consider the financial costs incurred by communities—prioritizing inexpensive PA options that can be sustained by communities beyond the conclusion of a study intervention or organizational program.

To the best of the authors’ knowledge, this study is the first in the literature on barriers and facilitators to use the PASSS [67] to assess all five established forms of social support. The findings from this study suggest that the LLPA group had less informational (i.e., having easy access to PA-supportive information and knowing how to use it) and instrumental (i.e., having the necessary resources and support to carry out PA) support than the HLPA group. The lack of informational support may be partially attributed to the lack of health literacy among the participants. Studies on the association between health literacy and PA participation have shown that high levels of health literacy are associated with higher levels of PA, and vice versa [133]. Regarding discrepancies in instrumental support, this may be attributed to minorities having fewer material or economic resources to lend others in their social circles to participate in PA. Considering the limited amount of work carried out to facilitate instrumental and informational support for this demographic, future PA programs may need to develop novel approaches to facilitate these forms of social support to promote regular PA.

Both study groups reported in their qualitative responses that they received social support for being physically active from their immediate family and extended social circles. These data are partially supported by the outcomes from the PASSS emotional support subdomain composite score, which demonstrated similar scores between the two groups. This finding is consistent with previous work suggesting that emotional and companionship forms of social support are key facilitators of PA among minorities [40,121,134]. Randomized control trials among minorities have consistently shown that emotional and companionship support can lead to higher rates of PA and increased enjoyment [55,68,69,135]. Together, these results further support the key role that emotional and companionship support can play in helping individuals to remain active.

Responses to the qualitative set of survey questions revealed that a lack of community PA facilities and public transit options was a barrier for both groups. Conversely, both groups reported that having more PA facilities/events and public transit options in their community would facilitate PA. Results from the PABQ revealed that the LLPA group perceived more physical environmental barriers than the HLPA group. Having close proximity and easy access to PA facilities (i.e., gyms, outdoor fields, parks) has been reported to facilitate PA for other regional minority populations in the United States [52,121,136]. The built environment is a key SDOH that has been shown to have a direct impact on health outcomes [6,137]. A recent panel study further supports this assertion, demonstrating that residents are more likely to report having good health status (i.e., no incidence of chronic disease, meeting the USPA guidelines) if their neighborhood environments have parks within walking distance, have higher walkability and biking scores, and have instituted Complete Streets policies [78]. Built environment policies such as Complete Streets [76] or zoning laws that dictate where community resources can be distributed play a pivotal role in community-level PA trends. Considering that minorities are more likely to reside in communities that lack PA resources and are more hazardous to pedestrians [34,138], more action must be taken by both investigators and legislators to provide PA-supportive environments for these communities.

### 4.1. Limitations and Strengths

This study has several strengths and limitations worth noting that aid in contextualizing these data. This project was executed using a cross-sectional study design, indicating that these data should be interpreted as associative rather than causal. The present study’s small sample size limits the generalizability of our findings and reduces the statistical power of our quantitative analysis. To address these constraints, we reported effect sizes alongside traditional significance testing. Effect sizes estimate the magnitude of the effect between two variables and are resistant to sample size influence [93,94]. Given this information, researchers should interpret these results with appropriate caution, recognizing the potential constraints imposed by this study’s small sample size.

It is also important to note the community and societal context in which this study’s participants reside (i.e., an urban Midwestern county with a growing minority population). In turn, these community and societal characteristics may limit the transferability of these findings to communities that share similar characteristics. The use of convenience and snowball recruitment for this study may have resulted in selection bias, wherein individuals with certain characteristics (i.e., those already motivated to be active; those inclined to participate in research studies) may be overrepresented. To mitigate against selection bias, we implemented several recruitment strategies, including leveraging community partners outside of the health and wellness space (i.e., local business), maintaining detailed documentation of recruitment chains, and monitoring demographic characteristics throughout data collection to identify potential gaps in community representation. Self-reported PA estimates have been historically documented to be prone to reporting bias [139] and overestimate PA compared to accelerometers. Careful attention was given to selecting a self-reported PA survey that has documented validity compared to accelerometers [102] and has wide transferability to different study populations [83].

Despite these limitations, this study advances the existing literature by presenting a comprehensive approach to exploring PA’s barriers and facilitators among minorities. Specifically, this study utilized a concurrent mixed-methods approach framed on the SEM, incorporating various scientific perspectives to answer the study’s specific aims. The present study also presented estimates of all four domains of PA, providing a complete picture of the participants’ weekly activity levels. Finally, this project incorporated numerous widely accessible, validated surveys that enhance its comparability with future studies and improve its methodological reproducibility.

### 4.2. Future Directions

Future research should build on the strengths of this study and address its limitations to advance our understanding of the factors influencing PA participation among minority adults. Specifically, the congruent quantitative and qualitative findings suggest the need for longitudinal investigations with larger sample sizes. These studies would benefit from using both objective devices and self-report measures to assess PA, as they provide complementary data. Moreover, studies carried out to understand PA’s barriers and facilitators among minority adults have largely been composed of female participants. Follow-up work should aim to address the limited data on the factors that prevent and promote PA among minority men. Lastly, future work should aim to replicate this work with a larger sample size and expand it into additional geographic (i.e., Mountain West, Pacific Northwest, etc.) and community contexts (i.e., rural) within the United States.

## 5. Conclusions

The findings from this study emphasize the need for health researchers and professionals to consider how factors across the SEM prevent and support PA among minorities. The findings from this study suggest that minorities completing low levels of leisure-time PA require a set of individual-level determinants to be present, such as motivational messaging, assistance in forming a PA habit, and planning PA into their schedules. These data also suggest that social support and accountability are key factors that help individuals engage in PA. Most notably, these data demonstrate that the financial cost of PA is a key concern for this demographic, stressing the need for community organizations and research teams to offer PA options that are free or at a significantly reduced cost to communities. These offerings should be provided as close to the communities of interest as possible, and they should aim to provide participants with various PA options that can be adapted to busy schedules. Taken together, these data can aid investigators and community stakeholders in constructing PA programs that address the key barriers faced by minorities.

## Figures and Tables

**Figure 1 ijerph-22-00234-f001:**
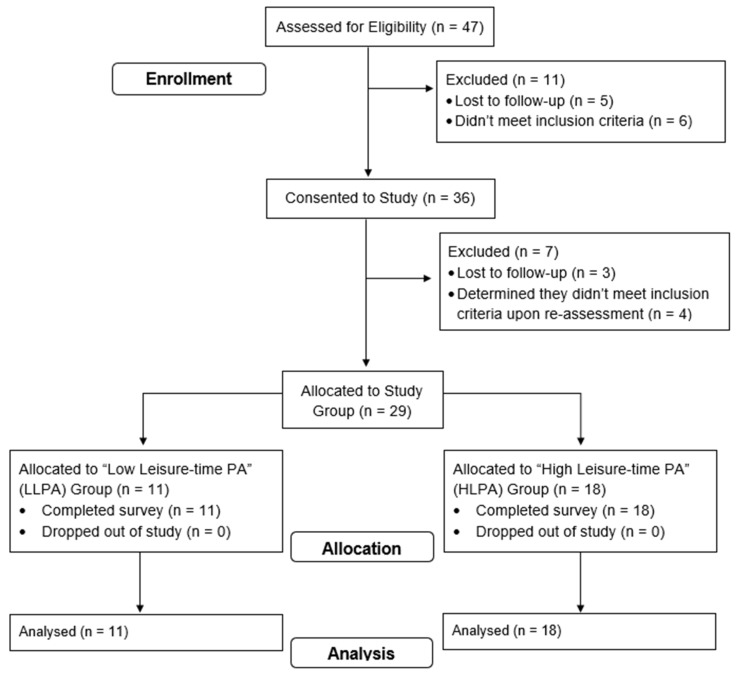
Participant study flow diagram: Forty-seven underrepresented racial minority adults (≥ 18 years) living within an urban Midwestern county were screened for eligibility, with twenty-nine participants consenting to and completing the study. The participants were recruited using various methods, including social media, email newsletters, and word of mouth.

**Table 1 ijerph-22-00234-t001:** Participant demographics: Twenty-nine community-dwelling underrepresented racial minority adults participated in this study. All data are self-reported. Data are reported as the mean (SD) where available.

Variable	LLPA	HLPA	Percent Difference	*p*-Value	Cohen’s *d*
Gender (n)					
Male	1.00	3.00	-	-	-
Female	10.00	15.00	-	-	-
Race/Ethnicity (%)					
Black	63.63	33.33	-	-	-
Hispanic/Latino	5.55	33.33	-	-	-
Asian	27.27	27.28	-	-	-
Middle Eastern	0.00	5.66	-	-	-
Education (% w/Bachelors or More)	54.45	72.22	-	-	-
Employed Full Time (%)	54.45	77.78	-	-	-
Personal Income (% > $50,000/yr.)	9.10	33.33	-	-	-
Did Not Answer (%)	18.18	16.67	-	-	-
Age (yrs.)	39.09 (18.21)	38.44 (14.15)	1.68	0.92	−0.04
Body Mass (kg)	80.95 (16.13)	73.23 (13.60)	10.01	0.18	−0.53
Height (cm)	165.45 (8.45)	164.11 (7.94)	0.81	0.67	−0.16
BMI (kg/m^2^)	29.75 (6.24)	27.32 (5.54)	8.52	0.29	−0.42
Years in Community	7.75 (8.81)	10.18 (11.76)	27.11	0.58	0.22
Total Leisure-Time PA (min/week)	44.55 (48.40)	448.89 (227.43)	163.89	0.001 *	2.21

* *p* < 0.05; BMI = body mass index; PA = physical activity.

**Table 2 ijerph-22-00234-t002:** Self-reported domain-specific physical activity outcomes and effect sizes: Values were computed from the International Physical Activity Questionnaire-Long Form delivered to participants as part of the study survey. All data are reported as minutes per week. Data are reported as the mean (SD).

Variable	LLPA	HLPA	Percent Difference	*p*-Value	Cohen’s *d*
**Occupational PA**					
MVPA	169.55 (372.90)	468.89 (903.25)	93.77	0.31	0.40
Walking PA	95.45 (94.70)	262.28 (379.35)	93.27	0.17	0.54
Total PA	265.00 (412.98)	731.17 (1062.74)	93.59	0.18	0.53
**Transportational PA**					
MVPA	0.00 (0.00)	10.00 (30.87)	200.00	0.30	0.41
Walking PA	140.91 (178.81)	178.56 (147.37)	23.57	0.54	0.24
Total PA	140.91 (178.81)	188.56 (164.30)	28.93	0.47	0.28
**Household MVPA**	284.55 (327.63)	273.78 (208.65)	3.86	0.91	−0.04
**Leisure-time PA**					
Exercise MVPA	40.00 (48.11)	238.89 (182.78)	142.63	0.00 *	1.34
Walking PA	4.55 (12.14)	210.00 (143.69)	191.52	<0.001 *	1.80
Total PA	44.55 (48.40)	448.89 (227.43)	163.89	0.001 *	2.21
**Summed Domain Totals**					
Total MVPA	494.09 (398.60)	991.56 (883.55)	66.97	0.09	0.67
Total Walking PA	240.91 (196.45)	650.83 (491.01)	91.94	0.01 *	1.01
Total PA	735.00 (521.94)	1642.39 (1073.98)	76.33	0.02 *	1.00

* *p* < 0.05; PA = physical activity; MVPA = moderate-to-vigorous physical activity.

**Table 3 ijerph-22-00234-t003:** Multivariate analysis summary table for survey composite scores: All survey data are expressed numerically. Survey questions for the PA Barrier Questionnaire, Self-Reported Behavioral Automaticity Scale, and PA Self-Efficacy Scale were delivered on a five-point Likert scale, with 1 = strongly disagree and 5 = strongly agree. Survey questions for the PA Social Support Scale were delivered as Likert scale items on a seven-point scale; with 1 = never true and 7 = always true.

Composite Score	Group	Mean (SD)	dƒ	MS	F-Statistic	*p*-Value	Cohen’s *d*
**PA Barrier Questionnaire (α = 0.92)**							
Personal	** LLPA **	36.64 (12.53)	10.00	556.14	5.72	0.02 *	−0.92
	** HLPA **	27.61 (7.88)					
Social Environment	** LLPA **	11.18 (3.57)	10.00	34.18	2.758	0.11	−0.64
	** HLPA **	8.94 (3.49)					
Physical Environment	** LLPA **	12.91 (5.52)	10.00	27.87	1.26	0.27	−0.43
	** HLPA **	10.89 (4.14)					
**Exercise Identity Scale (α = 0.89)**	** LLPA **	27.45 (7.90)	10.00	141.07	2.79	0.11	0.64
	** HLPA **	32.00 (6.61)					
**Self-Reported Behavioral Automaticity Scale (α = 0.97)**	** LLPA **	10.18 (5.06)	10.00	72.68	3.19	0.09	0.68
	** HLPA **	13.44 (4.60)					
**PA Social Support Scale (α = 0.84)**							
Emotional Support	** LLPA **	15.73 (4.58)	10.00	14.15	0.53	0.47	0.28
	** HLPA **	17.17 (5.48)					
Validation Support	** LLPA **	11.09 (5.80)	10.00	1.96	0.70	0.79	−0.10
	** HLPA **	10.56 (4.96)					
Informational Support	** LLPA **	13.18 (4.19)	10.00	34.95	1.15	0.29	0.41
	** HLPA **	15.44 (6.16)					
Companionship Support	** LLPA **	11.82 (3.71)	10.00	6.62	0.22	0.64	−0.18
	** HLPA **	10.83 (6.32)					
Instrumental Support	** LLPA **	7.18 (6.82)	10.00	231.12	3.79	0.06	0.75
	** HLPA **	13.00 (8.33)					

* *p* < 0.05; PA = physical activity.

**Table 4 ijerph-22-00234-t004:** Barriers to physical activity described by study participants: Responses were prompted by asking participants to answer questions from the Barrier Analysis Qualitative Survey. Participants were instructed to be as thorough as possible when answering each question. Identified themes, sub-themes, and quotes are organized by socio-ecological model (SEM) domains. Quotes are shown verbatim, with added clarification (denoted by brackets) to enhance readability and ease of interpretation.

Group	SEM Domain	Themes and Sub-Themes	Quotes
** LLPA **	**Personal**	Lack of Time	“Online classes [take up too much time]”
			“Work schedule [takes up too much time]”
			“[Physical activity] takes time away from work, causing me to give up sleep, take an unpaid lunch break, and/or work longer hours”
			“Balancing full time work and part time school [makes it hard to be active]”
		Physical Capacity	“[I have] low energy, lack of high workload,”
		Lack of Motivation	“Lack of motivation to be physically active [makes it difficult to be active].”
			“Takes a lot of time to see the results. [I] need to be motivated internally to go out and be active.”
		Physical Sensations Associated with PA	“Body pains after the exercise [make it difficult to be active]”
			“Weightlifting and other vigorous activity creates soreness.”
			“Sweating [makes activity difficult]”
			“Taking extra showers that can sometimes lead to dryer skin”
		Self-Doubt	“[I discourage] myself [from being active]”
		Cost Associated with PA	*Prevalent Sub-Theme in Community domain themes*
		Stress	“[I] worry about day to day stuff [like] bills”
		Risk of Injury	“Injuries can occur when active if not careful.”
	**Interpersonal**	Missing Out on Social Events	“[Physical activity] can become one of the reasons to skip some social events.”
	**Community**	Inclement Weather	“Not having any reason to go outside of my home. Bad weather like snowing/raining/strong winds.”
			“Some days, the weather doesn’t allow for me to get outside and enjoy walking or driving to the local YMCA for group exercise.”
		Lack of PA Facilities and Public Transit in Community	“[Not having] easy access to public transport [makes it harder to be active].”
** HLPA **	**Personal**	Lack of Time	“School… discourages me from being active as [physical activity] takes up too much time in my schedule.”
			“Family obligations, work obligations, and lack of time [make it hard to be active]”
			“When I miss a day. When I don’t get enough sleep. When I go out of town.”
			“Finding time. I have two jobs that take up every single day all day.”
			“Making time to do it and giving up other things that may be important [is a drawback of being active]”
		Lack of Time	“Work schedules and requirements that limit your time or drain you physically [discourage me from being active]”
		Lack of Motivation	“[Physical activity is] not easy, because it’s hard to get motivated”
			“I don’t have the energy or the motivation [to be active]”
			“Lack of motivation, lack of energy and lack of knowledge [makes it hard to be active]”
		Pregnancy	“Being pregnant [makes it hard to be active], as time goes on once baby is here will be finding daycare”
			“My mother [doesn’t support me being active] due to being almost 7 months [pregnant]”
		Risk of Injury	“Possible risk for injury if you don’t know what you’re doing or don’t have the correct form or stretch routine [is a drawback of being active]”
			“Aches and injuries [are drawbacks to being active]”
			“If you do not have proper training, you can do an exercise or lift a weight incorrectly causing harm to your body.”
		Physical Sensations Associated with PA	“{Being active] gets tiring and sometimes overwhelming”
			“Feeling lethargic, reduced mental clarity, and sore all the time [is a drawback of being active]”
		Cost Associated with PA	“Programs that are expensive and not local [discourage me from being active]”
			“Expensive gym memberships [discourage me from being active]”
			“It’s expensive [to be active]”
		Lack of Knowledge About PA	“[My] lack of knowledge on what [to] do in gym [discourages me from being active]”
	**Interpersonal**	Lack of Social Support	“Being active alone [discourages me from being active]”
	**Community**	Lack of PA Facilities and Public Transit in Community	“{Not having] a convenient gym [makes it hard to be active]”
			“Distance and transportation [to facilities discourage me from being active]”
	**Societal**	Social Norms Around Pregnancy	“Pregnant women shouldn’t be active”
		Cultural Norms Around Autonomy and/or Independence	“A cultural belief that is often shared by people in my religion is that parents are always in control of their children, no matter what their age is. They can somewhat discourage me being active if they say that we have to do something on the day that I would like to be active.”

**Table 5 ijerph-22-00234-t005:** Facilitators of physical activity described by study participants: Responses were prompted by asking participants to answer questions from the Barrier Analysis Qualitative Survey. Participants were instructed to be as thorough as possible when answering each question. Identified themes, sub-themes, and quotes are organized by socio-ecological model (SEM) domains. Quotes are shown verbatim, with added clarification (denoted by brackets) to enhance readability and ease of interpretation.

Group	SEM Domain	Themes and Sub-Themes	Quotes
** LLPA ** **PA**	**Personal**	Knowledge of PA Benefits	“The activities that is done to stay fit and it is very important in one’s health”
			“It keeps the body functions in better condition.”
			“It improves both mental and physical health including decreasing anxiety and depression, reducing conditions heart conditions, improve blood sugar levels, and improve cognitive and mental function.”
			“Physical activity leads to good cardiac [health] and joint mobility.”
			“It is like the elixir”
		Obtaining Health Benefits **Sub-Themes:** Physical Health, Mental Health, Emotional Health	“[Being active] helps one stay fit” “Being physically active helps me stay mentally fit as well. Helps me feel good about myself. If I workout in the morning, I tend to make better eating decisions as I am aware of all the hard work I have done in the morning.”
		Obtaining Health Benefits **Sub-Themes:** Physical Health, Mental Health, Emotional Health	“I sleep better and keep a healthy weight”
			“Better bone and muscle health. You don’t seem to age.”
			“Mental health benefits, more energy, increased confidence [are benefits of being active]”
			“It helps with sleep, mood, concentration, and balancing mental health struggles”
		Beliefs About the Body’s Value	“The fact that my body is my home, my safe space and I want to take care of it [encourages me to be active].”
		Goal Setting	“Hav[ing] a goal [helps me be more active]”
	**Interpersonal**	Social Accountability	“I [am encouraged to be active because I] have a service dog who needs to go on walks at least 2×/day”
			“[Having a] group of people committed to exercising [encourages me to be active]”
		Social Support	“Having a gym buddy [helps me stay active].”
			“My community consists of active older adults who are generally very physically active.”
			“My husband, my mother [support me to be active]”
			“My parents (specifically my dad) and some of my friends who are also physically active [support me]”
		Fostering Social Relationships	“Being active allows for social interactions with people in and around the gyms, pool, pickleball court.”
		Community PA Facility Availability	“[Having the] gym being open 24 h [help me be active].”
			“This summer I have a student rec membership, so I have access on the days I’m at campus.”
			“Inexpensive gyms within short walking distance of my job. Gym showers… The proximity of the canal.”
			“Regular cricket games in softball field. Having some table tennis tournaments.”
	**Community**	PA Community Events or Programs **Sub-Theme:** Cost of PA	“Regular cricket games in softball field. Having some table tennis tournaments.”
			“More sports tournaments/events. Having sport groups like TT group etc. Starting some learning sessions”
			“Free community offered classes, and physical events (yoga, dance, walking/running,). A community gym offered at a free and or very reduced rate.”
			“Silver Sneakers and YMCA Fitness Centers [help me be active]”
		Supportive Environments for PA	“Gym, less busy roads for walks, closer outlets [would help me be active]”
			“Free access to gym/ recreational activities and resources for beginners [would help me be active]”
			“Having access to [a] swimming pool, gymnasium, walking parks, bicycle paths”
		Age Appropriate PA Opportunities	“More resources for my age group [would help me be active]”
	**Societal**	Policies that Support and/or Subsidize PA (work, school, etc.)	“[Having access to] work sponsored programs. Canal sponsored programs [would help me be active].”
** HLPA **	**Personal**	Having a Goal and Routine	“I now have a routine. I do it in the morning before the day begins. I feel better when I exercise and sluggish when I don’t.”
		Enjoyment of PA	“[Physical activity is] something that genuinely makes me happy allowing me to enjoy it.”
			“Remembering the benefits of being physically active [helps me be active]”
		Knowledge of PA Benefits	“Sitting is the new smoking. In other words, you need to keep moving to stay healthy.”
			“I know that, for starters, physical activity doesn’t just play a role in physical health. It also plays a part in a person’s mental health.”
			“Physical activity is good for strengthening your heart and body.”
			“[I am] fairly knowledgeable [about the benefits of physical activity]. The more active you are, the better health you’ll have”
			“If you don’t use it, you will lose it. Staying active activates the happy hormone and relieves stress; it produces clarity of the mind, walking outside provides access to attaining vitamin D naturally and walking in nature produces some peace and grounding. Weightlifting helps combat bone loss and osteoporosis. Physical activity is providing a holistic measure for producing good health, mind, body, and spirit; good health is wealth.”
		Resiliency	“I really do not care about what others think of me, but most people who knows me knows that I have always been pretty active: family, friends, colleagues.”
		Obtaining Health Benefits **Sub-Themes:** Physical Health, Mental Health, Emotional Health Positive Emotional Health	“I notice the difference in my sleep, endurance, attitude, have more energy, helps with weight loss, healthy bones, mental stamina, bowel movements.” “Emotional and physical wellbeing [is improved with activity]” “Clear mind, good attitude, and decrease in depression [are benefits of being active]” “Better mental health, better mood, more energy and increases productivity.” “Improved sleep, mood, blood pressure and other health vitals. Decrease in obesity [are benefits].” “[Having lower] blood glucose level and [better] cardiovascular health”
			“Overall health would improve and physical appearance as well”
		Fitness Technology	“My Apple Watch keeps me accountable.”
			“Apple Fitness [encourages me to be active]”
		Having Resources Available to be Active	Having access to YouTube exercise videos, having a 10/month membership at planet fitness; I enjoy being active.”
	**Interpersonal**	Social Accountability	“Community or accountability partner [help me stay active].”
			“Being fit impresses people that you have self-respect”
		Social Support	“I do ‘walking Wednesdays’ with colleagues.”
			“Pretty much everyone around me encourages me to be active.”
			“I don’t think I have ever run into someone that has discouraged me from being active.”
			“My fiancé, kids, and best friend [support me to be active]”
			“Coworkers, roommates, parents, spouse [support me to be active]”
			“[Being active is a] way to make new friends.”
	**Community**	Community PA Facility Availability **Sub-Theme:** Cost of PA	“Having a gym at the property [makes it easier to be active]”
			“I work at a gym, and I have a gym in my apartment complex.”
			“Having an opportunity/space at work to work out for 30–60 min [would make it easier to be active].”
			“Exploring somewhere new and having a reason to leave the house”
			“More places to play tennis or pickleball [would help me be more active]”
			“Free gym memberships with a trainer [would help me be more active]”
			“Free access to gym/recreational activities and resources for beginners [would help me be more active]”
			“Free gym membership at work/on campus for staff [would help me be more active]”
			“[Having] a gym with a daycare nearby [would help me be more active]”
		PA Events or Programs	“One thing that [my employer] does is send out emails stating that there are activities taking place and other things like that. All these activities can help a person be more active.”
			“[My workplace wellness] program [encourage me to be active]”
			“Health and wellness promotion at [my job encourages me to be active]”
			“More diverse classes that are affordable [and] closer to home [would help me be more active]”
			“Being a [wellness program] ambassador [encourages me to be active]”
		Supportive Built Environments	“Where I work is conducive for walking and I do ‘walking Wednesdays’ with colleagues.”
			“I am fortunate to have sidewalks, and an inexpensive health club 5 min away.”
			“[Having facilities] close to my house, better sidewalks [would help me be more active]”
	**Societal**	Policies that Support and/or Subsidize PA (i.e., work, school, etc.)	“More work from home ability [would help me be more active]”
			“Decrease in insurance costs [encourage me to be active]”
			“I have Medicaid and they recently gave me a 6 month gym membership and a fitness tracker, so that’s a bit of encouragement”
			“Something with employment, money back for being active [would encourage me to be active]”
		Religious Beliefs Around the Body, Health	“In my religion, physical and mental health is emphasized and is stated that they are both incredibly important to a person. All of these things help me be more active.”
			“Take care of your body, your body is your temple”
			“Body positivity movement and the Bible says our bodies are temples for the Holy Spirit and that we should take care of them. 1 Corinthians 6:19–20 Also, the Daniel fast from the book of Daniel encourages healthy eating.”
		Knowledge of Racially Prevalent Health Conditions	“History of African Americans with high blood pressure and diabetes [encourages me to be active]”
			“I don’t want to end up with high blood pressure or diabetes”
		Cultural PA Norms	“Grew up with hard work and physical labor being a part of my life”

## Data Availability

The raw data supporting the conclusions of this article will be made available by the authors on request.

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
