# Peer review of "Comparing Barriers and Facilitators to Physical ActivityAmong Underrepresented Minorities: Preliminary Outcomes from a Mixed-Methods Study"

_ijerph, 2025, doi:10.3390/ijerph22020234_

Round 1
Reviewer 1 Report
Comments and Suggestions for Authors
The importance and necessity of the work should be clear in the introduction.
The sampling method is clear. In the results, the figure should be used and the significance of the figure should be clear. Mention the entry and exit criteria of the article. The limitations of the article should be mentioned. The reasons for improvement should be mentioned in the article
Author Response
Thank you for taking the time to review and provide feedback on our manuscript. Our responses to your feedback are outlined below. Any changes made to the manuscript based on your feedback are highlighted in yellow within the revised document.
Comment 1: The importance and necessity of the work should be clear in the introduction.
Response 1: Statements have been added to the end of Section 1.3 (lines 138 – 139; 153 – 156) that provide a clear explanation of the study’s importance and necessity. The same paragraph (lines 136 – 156) also highlights the current gaps in the literature that the study’s specific aims (Section 1.5) aim to address.
Comment 2: In the results, the figure should be used and the significance of the figure should be clear.
Response 2: We believe we have effectively and concisely reported our study outcomes in the Results section, using both in-text descriptions and the accompanying tables. Additionally, we have carefully reviewed the section to ensure that each figure and table is appropriately referenced within the text.
Comment 3: Mention the entry and exit criteria of the article.
Response 3: The inclusion and exclusion criteria for the study were originally presented in Section 2.2. Upon re-reading the section, we can see how the inclusion and exclusion criteria were unclear. Edits to the section have been made so that 1) The description of the inclusion and exclusion criteria starts a new paragraph, and 2) The inclusion and exclusion criteria are more clearly presented (lines 245 – 249).
Comment 4: The limitations of the article should be mentioned. The reasons for improvement should be mentioned in the article.
Response 4: We believe we have concisely outlined the key limitations of our study in Section 4.1 (lines 722 – 732). That said, we have added a “Future Directions” section (Section 4.2, lines 740 – 753) where we outline how future work can address the limitations of our study and provide a rationale for how these future directions can improve upon our work.
Reviewer 2 Report
Comments and Suggestions for Authors
I suggest to include percent differences in the comparison, is possible
Author Response
Thank you for taking the time to review and provide feedback on our manuscript. Our responses to your feedback are outlined below. Any changes made to the manuscript based on your feedback are highlighted in yellow within the revised document.
Comment 1: I suggest to include percent differences in the comparison, is possible
Response 1: It was unclear as to which portion of our dataset was referenced for this suggestion (i.e., demographics data, physical activity, survey measures, etc.). After discussion, we believed it was most appropriate to calculate percent differences for our continuous variables, including age, body mass, and self-reported PA. Those calculations were added to Table 1 and Table 2 were appropriate.
With that said, we have decided against calculating percent differences for our nominal (i.e., gender, race, education, etc.) and ordinal (i.e., survey composite scores derived from Likert Scale items) variables. Percent differences for nominal and ordinal variables can be easily misinterpreted, and don't represent true quantitative differences between variables. These calculations also don't lend themselves to standardized comparisons or further statistical analyses and would not provide additional context for the reader. Given these limitations, we believe that presenting our ordinal and nominal data as currently displayed is sufficient for interpretation by readers.
Reviewer 3 Report
Comments and Suggestions for Authors
The manuscript is well-supported by references, with good try on substantiating the finding by supporting literature. In section 1.2 consider adding references to broaden the discussion on the topic of environmental barriers and cultural influences on PA. Similarly, in section 1.3 address strategies that address these barriers with supporting literature. Make sure to spell out all abbreviations (i.e., practically significant effect (RMPE) abbreviations don’t match.
Author Response
Thank you for taking the time to review and provide feedback on our manuscript. Our responses to your feedback are outlined below. Any changes made to the manuscript based on your feedback are highlighted in yellow within the revised document.
Comment 1: In section 1.2 consider adding references to broaden the discussion on the topic of environmental barriers and cultural influences on PA.
Response 1: Thank you for this suggestion—these determinants are a great addition to the our broader discussion of health disparities and social determinants of health. Statements with supporting references have been added to Section 1.2 that discuss both broader environmental factors (lines 82 – 86) and culture (lines 88 – 91).
Comment 2: Similarly, in section 1.3 address strategies that address these barriers with supporting literature.
Response 2: While we understand the rationale behind including strategies to address existing barriers in this section, we believe that is beyond the scope of this section of our introduction. Our intention is to provide readers with an overview of key literature on the topic that helps provide basis and context for our investigation. With that said, the findings from our investigation provide corroborating evidence to many of the findings highlighted in this section. We explore these barriers in our discussion and provide numerous strategies and further lines of inquiry that could address these barriers (with supporting literature).
Comment 3: Make sure to spell out all abbreviations (i.e., practically significant effect (RMPE) abbreviations don’t match.
Response 3: Thank you for catching this discrepancy. We have opted to remove the RMPE abbreviation from the text and have replaced it with “practical effect size” as originally stated in Section 2.6.1. We have also removed the multiple imputation abbreviation (MI) from the text.
Reviewer 4 Report
Comments and Suggestions for Authors
This study provides a robust survey battery and post hoc qualitative analysis for less studied populations in the Midwest, which can provide data to help guide future interventions. Unfortunately, there are several limitations to this study, and not all are stated as limitations in the manuscript. More attention to explaining the methodology is required. There is valuable information gleaned from the study, but confidence in the findings is too low due to the limitations.
Introduction up until line 231
Line 194: Change the word “data” to results, findings, or something similar.
Line 195: I was expecting to see a paragraph on the organizational/institutional level of the social ecological model. Organizational level factors came up multiple times in the discussion and if the authors are going to lean into the implications for interventions based on this study, then the inclusion of organizational-level factors should be integrated into the paper.
METHODS: LINES 232 - 391
Line 240: When were participants randomized? How did they do random sampling by using existing partners, etc. Recruiting approaches identified describe convenience sampling more so than random sampling. The authors should clarify the randomization process and address the potential for selection bias.
Analysis
Lines 358-359: On line 153 and 154 in the introduction, the authors state that the study is critical for understanding how various factors currently influence PA participation. That sentence is written from a perspective that represents the supported causal relationship between barriers, facilitators, and physical activity behavior. That is barriers and facilitators are antecedents of behavior. For this reason, there is a strong argument that physical activity behavior should be the outcome or dependent variable in the analysis.
Line 376: Please clarify how many of the 29 participants provided extra statements for qualitative analysis and how many total extra statements. This information is crucial for understanding the scope and depth of the qualitative data.
RESULTS – LINES 393 – 566
Line 395: In the introduction differences in barriers and facilitators by sex was described as important and that most research had been conducted with women. This study includes 1 male in one group and 3 in the other (less than 14% of the sample in total). There is no distinction of qualitative data by sex.
Line 460: organizational level of social ecological model was not described in the introduction. The first qualitative barrier is then related to the limitation of time due to full-time work; makes me wonder about any research out there that investigate worksite factors that influence physical activity in general, and then more specifically, among minorities. Worksite/organizational level factors come up in other areas of the discussion.
Discussion
Line 641: A statement that future research should aim to enhance physical literacy in K-12 settings based on the results of a cross-sectional survey (with qualitative analysis) seems a bit of a stretch. Was the state (city) that this research was conducted have a de-prioritization in k-12 PA? Participants are 38-39 years of age. How relevant are the K-12 policy statements and this provides the years to see if the local school districts or state were deprioritizing K-12 PA. It makes sense to state that K-12 PA is the setting that could be in place to increase physical literacy, but the statement should be tempered.
Line 713-717: I understand that it isn’t the focus of the study, but with the focus of this study in one community with a small sample size and the stated limitations of generalizability even to communities that share similar characteristics as the community in this study, perhaps the more impactful research might be to identify and quantify the actual neighborhood and built environments and local policies and their relationship to physical activity among underrepresented people in the community.
Limitations
A limitation when investigating barriers and facilitators of physical activity in the Midwest is that data collection took place between January and June, which includes winter and summer – temperature, weather, etc. The authors could discuss the potential impact of seasonal variations in weather (temperature, snow, daylight hours) on physical activity levels and the study findings. How many participants completed the survey in the winter or late spring/summer?
Line 342: The a priori power analysis revealed needing a sample of 76 but ended up with 29. The authors should address the potential limitations of a small sample size, particularly regarding the generalizability of the findings and the statistical power of the analyses. It is unclear whether the inclusion of qualitative data mitigates the concerns associated with a small sample size. The authors should explicitly discuss this. While medium effect sizes may increase the likelihood of detecting practical significance, the authors should acknowledge the potential limitations of relying solely on effect size to address the concerns of a small sample size.
Recommendations
Line 743: It seems that future studies in this area need to replicate this study with a sample size that provides confidence of the outcomes before it is used as part of an intervention.
As a preliminary study the recommendations should primarily focus on what needs to be done to improve the current study. For example, findings suggest that physical limitations perhaps should be part of exclusionary criteria in future studies. How to increase sample size with this population (better recruit).
Author Response
Comment 1 - Line 194: Change the word “data” to results, findings, or something similar.
Response 1 - “Data” has been changed to “results” (line 194).
Comment 2 - Line 195: I was expecting to see a paragraph on the organizational/institutional level of the social ecological model. Organizational-level factors came up multiple times in the discussion and if the authors are going to lean into the implications for interventions based on this study, then the inclusion of organizational-level factors should be integrated into the paper.
Response 2 - The present study was developed using a four-domain version of the socio-ecological model previously used in the physical activity literature (Lee et al., 2013; Spence & Lee, 2002) and presently recommended by the United States Centers for Disease Control and Prevention (https://www.atsdr.cdc.gov/community-engagement/php/chapter-1/models-frameworks.html). That said, we appreciate you catching this oversight of our lack of mention of organizational factors in the introduction; and agree that these factors need to be mentioned given the findings of our study. We have revised section 1.4 to now include descriptions of organizational factors known to be associated with physical activity participation (lines 203 – 209).
Comment 3 - Line 240: When were participants randomized? How did they do random sampling by using existing partners, etc. Recruiting approaches identified describe convenience sampling more than random sampling. The authors should clarify the randomization process and address the potential for selection bias.
Response 3 - Line 240 has been revised to state “between January and June 2024, using convenience and snowball sampling.” We provide a detailed description of how participants were allocated to either study group in section 2.5 (Lines 329 – 338). We have included a statement in section 4.1 (Lines 727 – 734) regarding the potential for selection bias, and how we attempted to mitigate this within the present study.
Comment 4 - Lines 358-359: On line 153 and 154 in the introduction, the authors state that the study is critical for understanding how various factors currently influence PA participation. That sentence is written from a perspective that represents the supported causal relationship between barriers, facilitators, and physical activity behavior. That is barriers and facilitators are antecedents of behavior. For this reason, there is a strong argument that physical activity behavior should be the outcome or dependent variable in the analysis.
Response 4 - The sentence on lines 152 – 154 has been revised to suggest an associative relationship between external factors and physical activity participation. Given that our study is a cross-sectional investigation, causal interpretation cannot be inferred from our data and are limited to associations at best (Capili, 2022, Savitz & Wellenius, 2023). We have also made this clear in section 4.1 (lines 729 – 730). Psychological constructs known to be associated with physical activity behavior are included in our primary MANOVA analysis, and are reported in Table 3. Weekly estimates of physical activity have also been reported (Table 2).
Comment 5 - Line 376: Please clarify how many of the 29 participants provided extra statements for qualitative analysis and how many total extra statements. This information is crucial for understanding the scope and depth of the qualitative data.
Response 5 - All participants provide statements for the standardized set of qualitative questions (twelve questions in total) used in the study survey. This resulted in a total of 348 responses from all participants (132 statements for the LLPA group & 216 statements for the HLPA group). As mentioned in our methods section, the qualitative questions were given equal emphasis to the quantitative portion of the survey. The qualitative questions were placed towards the front of the survey (questions were presented right after the International Physical Activity Questionnaire) to prevent participant response fatigue.
Comment 6 - Line 395: In the introduction differences in barriers and facilitators by sex was described as important and that most research had been conducted with women. This study includes 1 male in one group and 3 in the other (less than 14% of the sample in total). There is no distinction of qualitative data by sex.
Response 6 - Given our study sample, we have deemphasized the lack of research done on men for this topic from the introduction (section 1.3). That said, we bring this gap in the literature to attention in section 4.2 (lines 759 – 762) and suggest exploring barriers and facilitators to physical activity among minority men as a future direction. When reporting our qualitative findings, we gave reporting outcomes by sex consideration. However, given our small sample size and the fact that we only had four men across both groups, we opted to report our findings by group rather than sex.
Comment 7 - Line 460: organizational level of social ecological model was not described in the introduction. The first qualitative barrier is then related to the limitation of time due to full-time work; it makes me wonder about any research out there that investigates worksite factors that influence physical activity in general, and then more specifically, among minorities. Worksite/organizational level factors come up in other areas of the discussion.
Response 7 - We have revised the introduction of the manuscript to include describing organizational factors and their association with physical activity participation. To your point, there is some literature published on the association between worksite environment and physical activity. We cite one such instance in the introduction, and other literature related to this topic are cited in the discussion.
Comment 8 - Line 641: A statement that future research should aim to enhance physical literacy in K-12 settings based on the results of a cross-sectional survey (with qualitative analysis) seems a bit of a stretch. Was the state (city) that this research was conducted have a de-prioritization in k-12 PA? Participants are 38-39 years of age. How relevant are the K-12 policy statements and this provides the years to see if the local school districts or state were deprioritizing K-12 PA. It makes sense to state that K-12 PA is the setting that could be in place to increase physical literacy, but the statement should be tempered.
Response 8 - We have removed the statement regarding physical literacy and K-12 policy, and have replace it with a statement calling for future work to develop feasible approaches to enhancing physical literacy in adults (lines 640 – 642).
Comment 9 - Line 713-717: I understand that it isn’t the focus of the study, but with the focus of this study in one community with a small sample size and the stated limitations of generalizability even to communities that share similar characteristics as the community in this study, perhaps the more impactful research might be to identify and quantify the actual neighborhood and built environments and local policies and their relationship to physical activity among underrepresented people in the community.
Response 9 - We agree with you on this, and are actively developing future studies that will build off the present study that will identify and describe our community’s built environment and existing physical activity-supporting policies.
Comment 10 - A limitation when investigating barriers and facilitators of physical activity in the Midwest is that data collection took place between January and June, which includes winter and summer – temperature, weather, etc. The authors could discuss the potential impact of seasonal variations in weather (temperature, snow, daylight hours) on physical activity levels and the study findings. How many participants completed the survey in the winter or late spring/summer?
Response 10 - While recruitment for the study began in January 2024, the first wave of participants was not enrolled until early March, with the first participants completing the study after March 21, 2024 (early Spring). Participant enrollment continued through the end June 2024 (early Summer). Of the 29 participants included in the sample, four completed the study in March; 15 completed in April; two completed in May; and eight completed in June. According to records from the U.S. National Weather Service (https://www.weather.gov/wrh/climate), average temperatures during this time period were above average for the region, yielded below average precipitation, and produced no severe weather events. In our sample of participants, inclement weather did not appear to be a prevalent barrier to physical activity. Our qualitative analysis revealed that only two participants in the LLPA group mentioned inclement weather as a barrier (see Table 4), and no mentions were made by participants in the HLPA group. The question in the PA Barrier Questionnaire explicitly asking about bad weather did not yield differences between the two groups. In turn, we have opted to not discuss inclement weather in our discussion of our findings.
Comment 11 - Line 342: The a priori power analysis revealed needing a sample of 76 but ended up with 29. The authors should address the potential limitations of a small sample size, particularly regarding the generalizability of the findings and the statistical power of the analyses. It is unclear whether the inclusion of qualitative data mitigates the concerns associated with a small sample size. The authors should explicitly discuss this. While medium effect sizes may increase the likelihood of detecting practical significance, the authors should acknowledge the potential limitations of relying solely on effect size to address the concerns of a small sample size.
Response 11 - We have included a statement in section 4.1 (lines 730 – 736) the address the limitations of our small sample size and the use of effect size differences to interpret our quantitative study findings. We do not see the qualitative dataset as trying to mitigate the limitations of the small sample size. In concurrent mixed methods research, qualitative data is complementary to the quantitative and helps the investigative team provide a more complete picture of the topic under study (Tashakkori & Teddlie, 2010).
Comment 12 - Line 743: It seems that future studies in this area need to replicate this study with a sample size that provides confidence of the outcomes before it is used as part of an intervention.
Response 12 - We have added the statement in section 4.2 (line 757) to acknowledge the need to replicate this study with a larger sample size.
Round 2
Reviewer 1 Report
Comments and Suggestions for Authors
edited and its can be publish
Author Response
Comment: edited and its can be publish
Response: We appreciate the time you took to provide feedback on our work. Thank you!